# Regional Disparity and Patients Mobility: Benefits and Spillover Effects of the Spatial Network Structure of the Health Services in China

**DOI:** 10.3390/ijerph18031096

**Published:** 2021-01-26

**Authors:** Liping Fu, Kaibo Xu, Feng Liu, Lu Liang, Zhengmin Wang

**Affiliations:** 1College of Management and Economics, Center for Social Science Survey and Data Tianjin University, Tianjin 300072, China; lpf3688@126.com (L.F.); lianglu96@tju.edu.cn (L.L.); jormongandr@tju.edu.cn (Z.W.); 2Politics and Public Administration College, Qinghai Nationalities University, Xining 810007, China; 3School of Public Finance and Administration, Tianjin University of Finance & Economics, Tianjin 300222, China; liufeng@tjufe.edu.cn

**Keywords:** health economy, spatial network, health inequity

## Abstract

Background: The distribution of medical resources in China is seriously imbalanced due to imbalanced economic development in the country; unbalanced distribution of medical resources makes patients try to seek better health services. Against this backdrop, this study aims to analyze the spatial network characteristics and spatial effects of China’s health economy, and then find evidence that affects patient mobility. Methods: Data for this study were drawn from the *China Health Statistical Yearbooks* and *China Statistical Books*. The gravitational value of China’s health spatial network was calculated to establish a network of gravitational relationships. The social network analysis method was used for centrality analysis and spillover effect analysis. Results: A gravity correlation matrix was constructed among provinces by calculating the gravitational value, indicating the spatial relationships of different provinces in the health economic network. Economically developed provinces, such as Shanghai and Jiangsu, are at the center of the health economic network (centrality degree = 93.333). These provinces also play a strong intermediary role in the network and have connections with other provinces. In the CONCOR analysis, 31 provinces are divided into four blocks. The spillover effect of the blocks indicates provinces with medical resource centers have beneficial effects, while provinces with insufficient resources have obvious spillover effects. Conclusion: There is a significant gap in the geographical distribution of medical resources, and the health economic spatial network structure needs to be improved. Most medical resources are concentrated in economically developed provinces, and these provinces’ positions in the health economic spatial network are becoming more centralized. By contrast, economically underdeveloped regions are at the edge of the network, causing patients to move to provinces with medical resource centers. There are health risks of the increasing pressure to seek medical treatment in developed provinces with abundant medical resources.

## 1. Introduction

As the largest developing country in the world, China has experienced rapid economic development and rapid improvement in medical care in the past 40 years. In 2009, China began a new round of health care system reform. The goal of the reform is to solve the problem of difficult and expensive medical treatment for patients for a long time. According to the goal of medical and health system reform, the government strengthens the public health care system and establishes a basic medical security system covering urban and rural residents. The government carries out hierarchical diagnosis and treatment in public hospitals to make full use of medical resources. However, health inequality among its citizens is on the rise, mainly as a result of the unreasonable distribution and unfair allocation of health resources. Therefore, research on the spatial correlation of healthy economic development is of great significance for the formulation and implementation of health policies.

In recent years, scholars have studied the spatial correlation structure of a healthy economy from the perspective of spatial structure evolution, spatial difference analysis, and its influencing factors [1,2,3,4]. Regional differences allow governments in different regions to spend different amounts on health care [5]. Therefore, the difference in the health economy is that the gap between regional medical service capabilities is increasing. Due to the differences in economic development among the eastern, central, and western regions, health resource allocation and health service utilization in China is inequitable and inefficient [6]. The empirical results show that the regional distribution of medical institutions, medical staff, and beds in China is extremely uneven, with the Gini coefficient exceeding 0.7 [7]. In addition, in terms of primary health care and urban and rural medical services, there are significant differences in different regions [8]. Some theorists have suggested that the “economic status of cities accounted for most of the existing inequality” [9]. Empirical results show that “most of this inequality is accounted for by within-province inequalities (82% or more) rather than by between-province inequalities” [10]. Although many studies have explained the unreasonable status of medical resources allocation in China [3,6], most of which do not analyze the spatial structural characteristics of the health economy. China’s healthy economic development is indeed uneven between regions and cities, but it has both mobility and spillover in space. Future research should start from the spatial structure of the health economy and rationally allocate medical resources according to the size and radius of patients served [11].

At present, the main methods to measure the allocation of health resources at the spatial level include provider-to-population ratios and shortest path analysis. Provider-to-population ratios are relatively traditional, but the mean population cannot effectively distinguish the difference in medical space and accessibility within the region [12], nor does it explain the spatial accessibility of patients near regional geographic boundaries [13]. The shortest path analysis is represented by linear distance or transportation network distance. However, this method may be more suitable for rural areas than cities [14]. The gravitational model takes into account not only the decrease in accessibility with the increase in transportation cost but also the medical service supply capacity of the medical service point [15]. The main research methods to study the imbalance and inequality of medical resource allocation are the Gini coefficient, Thiel index, concentration index, DEA, etc. [7,16]. However, these methods only discuss regional differences, and fail to further explain the regional interactions caused by regional differences. Therefore, the social network method provides a solution that can not only identify regional health and economic differences, but also further takes spatial effects into account.

This study uses 2017 data as a case study and takes 31 provinces in China—excluding Hong Kong, Macao, and Taiwan—as the basic analysis unit and uses the modified gravity model and social network analysis to verify the current spatial correlation state of China’s medical resources. This article focuses on the spatial elements that affect the layout of health service institutions and determines the current spatial pattern of China’s medical and health economy in terms of the benefits and spillover effects. The ultimate goal is to find the best spatial layout of medical service institutions, solve the problems of fairness and efficiency in the allocation of medical and health resources, and narrow the differences in the health conditions of residents in different regions. This study can make up for the deficiencies of the previous literature on the imbalance of medical services and supplement the methodology. Through the method of spatial network analysis, this paper explains the reasons for the flow of patients and evaluates the consequences of the imbalance of resources. The government has enough space to formulate health policies and reduce the cross regional flow of patients by balancing resources.

## 2. Method and Materials

### 2.1. Modified Gravitational Model

Referring to previous studies in the literature [17,18,19], this section mainly discusses the construction of a relational matrix through the modified gravity model, such as that in Formula (1). In the spatial correlation network of the regional medical economy, each region is a point in the network, and the health economic spatial relationship between regions is the line in the network. The points and lines together constitute the regional health economic spatial association network. Therefore, the regional health economic spatial association network establishes the relationship between each region. According to the existing literature, the construction of a health economic spatial correlation network mainly adopts the VAR Granger causality test method [20] and gravity model method [21]. However, the VAR method is too sensitive to data samples, which impacts the accuracy of the test results; thus, it has difficulty in accurately measuring the spatial correlation of the health economy. The gravity model can accurately measure the health economic attractiveness between regions and thus provides a scientific and reasonable method for determining the correlation of health economics within the region. This paper uses this method to solve the spatial correlation problem of the health economy. This method can not only directly observe the distribution characteristics of China’s medical spatial network but also comprehensively consider the influence of geographic distance on the overall network structure, which is conducive to ensuring the accuracy and scientificity of the measurement results. This paper argues that the reason for the attraction is the difference of medical resource allocation among different regions, including the number of tertiary hospitals, government health investment, economic development level, geographical distance, etc. Therefore, according to the above assumptions, the gravitational model is modified and the following model is obtained.
(1)$$Y_{ij}=K_{ij}\frac{\sqrt[{3}]{P_{i}H_{i\;}G_{i}}\sqrt[{3}]{P_{j}H_{j\;}G_{j}}}{{\left(\frac{D_{ij}}{g_{i}-g_{j}}\right)}^{2}}$$
(2)$$K_{ij}=\frac{R_{i}}{R_{i\;}+R_{j}}$$

In Formula (1), *i* and *j* represent different provinces. *K_ij_* represents the gravitational coefficient. *Y_ij_* represents the medical gravitation between the two provinces. *P_i_* and *P_j_* represent the number of visits. *H_i_* and *H_j_* represent the total health expenditure in two different provinces. *G_i_* and *G_j_* represent the GDP in two different provinces. *D_ij_* represents the distance between provincial capital cities in two different provinces. *g_i_* and *g_j_* represent the GDP per capita in two different provinces. (*g_i_ − g_j_*) represents the economic distance of different provinces. In Formula (2), *R_i_* and *R_j_* represent the number of tertiary public hospitals in two different provinces.

According to the calculation results of Formula (1), the attribute data are converted into a relation matrix as follows. First, the model calculates the gravitational value matrix ($M={\left(m_{ij}\right)}_{31\times 31}$) of the medical relationship between the two provinces, excluding Hong Kong, Macao, and Taiwan; second, from the average of the rows of the gravitational value matrix, a benchmark is established, as shown in Formula (3). If the gravitational value in the same row is higher than the assigned value of the benchmark, the value is 1, indicating that one province’s medical care is associated with the medical care of the other province. In contrast, if the gravitational value in the same row is lower than the assigned value of the benchmark, the value is 0, indicating that one province’s medical care is not associated with the medical care of the other province. According to the calculation results, the points in the whole network are connected to construct a complete and directional health economic spatial network structure.
(3)$$m_{ij}=\left\{\begin{array}{c} 0\;\;\;\;k_{ij}<\frac{\sum_{j=1}^{n}m_{ij}}{n}\\ 1\;\;\;k_{ij}\geq \frac{\sum_{j=1}^{n}m_{ij}}{n} \end{array}\right.$$

### 2.2. Social Network Analysis (SNA)

Social network analysis is a research method to study the relationship among actors. The actors can be people, communities, organizations, countries, etc. The major SNA indicators employed include network density, centrality, matrix correlation, etc. [22]. The SNA attempts to account for individual’s behaviors/characteristics by contemplating the social structure and the nature of social interactions among actors in a network [23]. As such, SNA explains the behavior and social process within a group through the connections among its members [24]. Studies with this approach often attempt to understand the structural properties of a network using network mapping and statistical measures that assess density and centrality. Thus, the social network approach has proven to be instrumental in exploring social phenomena in many disciplines [25].

#### 2.2.1. Overall Network Characteristics

The overall network characteristics of the inter-province medical economic analysis include the composition of the overall network according to the network relationship, including density, connectedness, hierarchy, and efficiency.
Network density reflects the degree of density of inter-province medical economy relationships. The calculation of network density is the ratio of the actual number of relationships between provinces to the maximum possible relationship of the overall network. The formula is D = $\frac{L}{N\left(N-1\right)}$. D is the network density, *L* is the number of relationships actually possessed, and *N* is the number of regions.Connectedness is used to measure the robustness of the medical economic spatial association network. The higher the correlation is, the stronger the robustness and the deeper the participation of the spatial association network in the medical fields in each province. The formula is C = $1-\frac{V}{N\left(N-1\right)/2}$. C is the connectedness, *V* is the logarithm of the unreachable point in the network, and *N* is the number of regions.Hierarchy is used to measure the extent to which the provinces in the network are asymmetrically reachable. The higher the network level is, the more stringent the network. A few provinces in the medical spatial network play a leading and dominant role. The formula is $H=1-\frac{K}{\max\left(K\right)}$**.** H is the hierarchy, *K* is the logarithm of the symmetrically reachable point in the network, and max (*K*) is the logarithm of the largest possible reachable point in the network.Efficiency reflects the connection efficiency between the provinces in the spatial network of medical economic development. The higher the network efficiency is, the fewer the connections between the provinces, the looser the spatial network, and the greater the medical resource flow in each province. It is difficult for inter-province collaboration and promotion in the medical field to be achieved through the network. The formula is $E=1-\frac{M}{\max\left(M\right)}$**.** E is the efficiency, *M* is the number of extra lines in the network, and max(*M*) is the maximum number of possible extra lines.

#### 2.2.2. Ego Network Characteristics

The ego network characteristics of the inter-province medical economy refer to the position of the actor in the network, including degree centrality, closeness centrality, and betweenness centrality.
Degree centrality reflects the central position of a single province in the medical economic spatial network. The higher the degree centrality is, the more connections there are with other provinces, and the more centrally located the province is in the network. The formula is $De=\frac{n}{N-1}$. *De* is the degree centrality, *n* represents the number of provinces directly associated with the province, and *N* represents the maximum number of connected provinces. The degree centrality includes two indicators: out-degree and in-degree. When the out-degree is greater than the in-degree, the node will exhibit a spillover effect; otherwise, it will have a beneficial effect. When the two are equal, there will be equilibrium.Closeness centrality reflects the degree of direct association of individual provinces with other provinces in the spatial association network of the medical economy. The higher the closeness is, the more direct inter-province contact there is; thus, the province is a central actor in the network. The formula is $C_{APi}^{-1}=\overset{n}{\underset{i-1}{\sum }}d_{ij}$. $C_{APi}^{-1}$ is closeness; *d_ij_* represents the shortcut distance between *i* and *j*.Betweenness centrality reflects the extent to which a province is in the middle of the path of other medical economics. In the spatial network structure of medical economic development, the higher the intermediary degree of a province is, the more relevant the province. On the shortest path, the stronger the ability to control the relationship between other provinces, the greater the role of the “center” or “bridge” played by other provinces. The formula is $Cb_{i}=\frac{2\sum_{j}^{n}\sum_{k}^{n}b_{jk}\left(i\right)}{N^{2}-3N+2}$ (j≠i≠k,j<k). *Cb_i_* is betweenness, and *b_jk_(i)* is the ability of the third region *i* to control the association of *j* and *k*.The block model analysis (CONCOR analysis) in social network analysis is used mainly to describe the roles and status of each province in the overall medical economic spatial network structure. To facilitate an intuitive analysis, the complex network can be simplified into a block model and image matrix. The spatial clustering method is used to divide the network into blocks. The block models reflect the role of each block in the network. Block models were proposed by White in 1976 [26] to analyze the role of network node groups. Referring to the indicators of Wasseman and Faust to evaluate the internal modules of the network [27], the regional medical economic spatial association network can be divided into four sections: the net beneficial block, bidirectional spillover block, net spillover block, and broker block. The members of the net beneficial block have more internal relationships, fewer external relationships, and fewer spillover effects on other blocks. The members of the bidirectional spillover block have more relationships with members of this block and members of other blocks but receive fewer external contacts and have a two-way spillover effect on members in other blocks. In the broker block, there are fewer connections between members within the block, and members of other blocks receive and issue relationships, acting as a “bridge” in the network. The members of the net spillover block have more relationships with members of other blocks and have fewer external relationships. Through CONCOR analysis, this study examines the inter-province medical economic spatial association network and the internal structure status of the block.

### 2.3. Data Collection

Data for this study, GDP, GDP per capita, and population come from the China Statistical Yearbook 2018. The number of outpatient visits, total health expenditure, bed utilization, and the number of tertiary public hospitals come from the China Health Statistics Yearbook 2018 and China Health Statistics Yearbook 2019. Based on the data, this article takes 31 provinces in China, excluding the Hong Kong, Macao, and Taiwan regions, as research objects, and empirically examines the spatial correlation of the interprovincial medical economy. The geographical distance between provinces is expressed by the spherical distance between provincial capitals and is measured by ARCGIS10.2 software (Environmental Systems Research Institute, Redlands, CA, USA).

## 3. Results

### 3.1. Overall Network Structural Characteristics of China’s Provincial Health Economic Spatial Network

According to the results of the modified gravity model, Figure 1 depicts the structure of the health economic space (the definitions of provinces and regions are given in Table A1 in the Appendix A). The number of spatial associations of China’s provincial health economics in 2017 is 209, and the provinces are indispensable in the spatial network structure of the health economy. China’s health economy does not have isolated provinces in space, and the provincial health economy has universal connectivity in the spatial economy. Table 1 depicts the structure of the network. The density of the network of the health economy is 0.2247, indicating that the degree of health economics between provinces is not high. The mobility of health resources between provinces is relatively low. Therefore, if there is a regional imbalance, health resources can be concentrated in a certain province. To obtain high-quality medical services, patients must have regional mobility.

The result of connectedness is 1. Thus, there is a direct or indirect health economic space spillover relationship between provinces. This feature is shown in Figure 1, that is, there is a general connection between the provinces, which indicates health economic relations. The hierarchy is 0.4541, and the level of the hierarchy is medium. Although the network structure of China’s interprovincial health economy is not completely hierarchical, there are some imbalances among the provinces. The efficiency is 0.6920, and the value of the efficiency is greater than 0.5. There is spatial mobility of patients between provinces, and the flow rate is relatively high, which indicates that due to the imbalance of resource allocation between provinces or the relatively static resource flow, patients choose to seek medical treatment in different places.

### 3.2. Ego Network Structural Characteristics of China’s Provincial Health Economic Spatial Network

#### 3.2.1. Degree Centrality

The results show that the highest and lowest values of degree centrality of the 31 provinces in the country are 93.333 and 16.667, respectively. The average degree centrality is 35.269. Degree centrality visualization is achieved using Geoda software (University of Chicago, Chicago, IL, USA) with the natural breaks method. Figure 2 depicts the distribution of degree centralities in different provinces. There are significantly more economically developed areas in the east than in other areas, which indicates that the imbalance of the health economy is still relatively obvious. The eastern region, especially Beijing, Tianjin, Shanghai, Jiangsu, and Zhejiang, has become the preferred medical treatment area for patients. The correlation between the health economies of 31 provinces across the country shows a high degree of medical treatment for these provinces. Several reasons explain these issues. First, due to the high level of economic development in the region, medical resources are relatively abundant there. In contrast, medical resources are relatively scarce in the central and western regions. Next, the gap in medical resources in different regions, such as talent and equipment, is the main reason why people seek medical treatment in different places. Finally, the eastern region’s better economic foundation, greater medical talent, more advanced medical equipment, greater number of tertiary medical institutions, and other factors all affect the attractiveness of the region’s health economy, and the rate of patient flow increases.

In terms of in-degree data, as shown in Table 2, Shanghai, Beijing, Tianjin, Zhejiang, Guangdong, and Shandong have high in-degrees, and they have become the main target provinces for medical treatment. At the same time, most of the central and western provinces have high out-degrees, and patients have a tendency to flow outwards. In terms of degree centrality rankings, the top five provinces are Shanghai, Jiangsu, Beijing, Tianjin, and Zhejiang. These five provinces have the most direct relationship with other provinces in the spatial association network structure of the health economy. Compared with other provinces, they are in a more central position in the network. The last six provinces are Jilin, Inner Mongolia, Hebei, Shanxi, Liaoning, and Ningxia. These six provinces have the lowest degree centrality, and they are in a subordinate position. From the perspective of benefits, 8 of the 31 provinces in the country are benefiting provinces, including Shanghai, Jiangsu, Beijing, Tianjin, Zhejiang, Shandong, Henan, and Anhui. Two provinces are balanced provinces, including Fujian and Hebei. The remaining 21 provinces are spillover provinces. However, it should be noted that benefits refer to the benefits caused by the flow of patients due to the attraction of medical resources, while the spillover effect is the opposite effect.

#### 3.2.2. Closeness Centrality

The average closeness centrality is 62.106. The provinces with the lowest values of closeness centrality are Liaoning and Ningxia (closeness centrality = 54.545), the provinces with the highest values are Shanghai and Jiangsu (closeness centrality = 93.750), and the closeness centralities of Beijing and Tianjin are both above 80. From Table 2, it can be seen that 31 provinces in China can quickly contact other provinces in the spatial network for medical resources, which verifies the spatial correlation. Due to the imbalance between the demand and allocation of patients’ health and medical care services, the probability of patients receiving medical treatment allopathically is gradually increasing. The resulting flow of people in various regions makes the health and medical connections in various regions increasingly close. Provinces above the mean value of closeness centrality are located in the eastern coastal area of China, including Shanghai, Jiangsu, Beijing, Tianjin, Zhejiang, Guangdong, and Shandong. These provinces with good economic development and rich health and medical resources, especially the cities of Beijing, Tianjin, Shanghai, and Jiangsu, have intimate connections with other provinces and can have important impacts. Meanwhile, the problem is that the reception pressure of these central actors is much higher than that of other provinces.

#### 3.2.3. Betweenness Centrality

It can be seen from Table 2 that the average betweenness centrality of 31 provinces is 2.232. The top ten cities are Shanghai, Jiangsu, Beijing, Tianjin, Zhejiang, Guangdong, Shandong, Fujian, Gansu, and Henan, with a total of 64.536, accounting for 93.318% of all betweenness centrality. The provinces with high control and influence play a strong bridging role in the spatial network structure. Most of the correlations between health and medical care in the whole network are achieved through these provinces with high betweenness centrality. Except for Gansu and Henan, the other eight provinces are located along the southeast coast and feature developed economies and convenient locations. Gansu and Henan are close to central areas of China and play the role of intermediaries to undertake floating patients in western provinces and other places. The ten cities with the lowest betweenness centrality are Xinjiang, Heilongjiang, Shaanxi, Hebei, Shanxi, Inner Mongolia, Liaoning, Jilin, Qinghai, and Ningxia, with a total of 0.077, accounting for 0.111% of all provinces. These cities are concentrated in the northeast area and the inland remote areas of China. These ten provinces are difficult to access geographically, so they have weak control in the network structure.

In short, from the perspective of centrality analysis, 31 provinces across the country have a general relationship, and the benefits and spillovers of individual provinces in the network are obvious. Their developed economies and obvious concentration of medical resources not only bear the important content of medical services but also play a strong intermediary role.

### 3.3. CONCOR Analysis

Based on the spatial correlation of the provincial health economy in China in 2017, using Ucinet software (University of California at Irvine, Irvine, CA, USA), an iterative analysis method was used to select a maximum segmentation degree of 2 and a convergence criterion of 0.2. The 31 provinces of China were divided into four sectors. China’s health economic development spatial network associations totaled 209, the number of internal relationships among the four sectors was 18, and the number of relationships among the sectors was 191, indicating that there is a clear health economic spillover relationship between the four blocks.

Combining Figure 3 and Table 3, Block 1 includes Beijing, Tianjin, and Shandong, which are developed provinces in North China. In Block 1, there are 18 total spillovers and 4 intrablock spillovers. The expected ratio of internal relations is 6.667%, and the actual ratio of internal relations is 22.222%; thus, Block 1 is the “net beneficial block”. The characteristics of this block indicate that provinces in it have accepted more patients from other provinces. In particular, these areas are not only more economically developed but are also rich in medical resources. Beijing and Tianjin, as the political and economic centers of the country, have a higher level of medical services than most regions in the country, which has caused more patients to flood into Beijing and Tianjin. Beijing has general hospitals, and Tianjin features specialist hospitals that attract a large number of patients looking for medical resources. For example, in 2017, the population of Beijing was only 18 million, but doctors were responsible for 9.3 consultations per day. In Tianjin, there are only 15.17 million people, but the number of daily consultations was 10.4. These two provinces rank among the highest in the country in terms of doctors’ daily consultations. The increase in the population outside the region has led to a sharp rise in pressure for medical treatment in Beijing and Tianjin.

Block 2 includes five provinces, Jiangsu, Guangdong, Fujian, Zhejiang, and Shanghai, which are distributed mainly along the southeast coast of China. There are 36 total spillovers in Block 2 and 8 intrablock spillovers. The expected ratio of the internal relation is 13.333%, and the actual ratio of the internal relation is 22.222%; thus, Block 2 is the “bidirectional spillover block”. The province locations determine the characteristics of the block. For example, the Yangtze River Delta radiates to other provinces, with Shanghai, Hangzhou, and other cities as the center, and the Pearl River Delta region radiates with Guangdong as the center. Fujian is in the connection zone between the Yangtze River Delta and the Pearl River Delta. Thus, in several provinces in Block 2, patients from other provinces may flow to these provinces, thus creating the possibility of spillover.

There are 12 provinces in Block 3, including Jilin, Hebei, Inner Mongolia, Shaanxi, Liaoning, Qinghai, Heilongjiang, Henan, Hubei, Shanxi, Ningxia, and Chongqing, which are mainly distributed north of the Yangtze River in China. There are 73 total spillovers in Block 3 and 5 intrablock spillovers. The expected ratio of the internal relation is 36.667%, and the actual ratio of the internal relation is 6.849%; thus, Block 3 is the “broker block”. These provinces have typical “intermediary” characteristics. For example, Chongqing is an important province that connects the southwest region, Hubei connects the north and south, Hebei is adjacent to Beijing and Tianjin, and Liaoning has become a connecting province in the northeast. These places are close to the core area of medical resources and feature many hospital branches. Therefore, patients can return to the province as soon as possible after treatment in a central city.

Block 4 includes Hunan, Hainan, Guangxi, Guizhou, Yunnan, Tibet, Anhui, Gansu, Jiangxi, Sichuan, and Xinjiang, which are mainly located in southwest and northwest China. There are 82 total spillovers in Block 4 and 1 intrablock spillover. The expected ratio of the internal relation is 33.333%, and the actual ratio of the internal relation is 1.219%; thus, Block 4 is the “net spillover block”. Most of the provinces in this block belong to less-developed regions and are far from developed provinces. The main reasons for this phenomenon are as follows: on the one hand, there is a relative lack of local medical resources and professional medical service personnel, especially for patients with major diseases who can no longer obtain effective treatment locally and have to be transferred to eastern regions with rich medical resources. On the other hand, some people, especially retirees, relocate to obtain better medical and health services. Therefore, patients have flowed to areas with developed medical resources.

To examine the relationship between the health economy among the blocks and reflect the distribution of health economic spillovers in each section according to the density criterion, the density between the various blocks is calculated first. Then, if the density is greater than the overall network density, the value is 1, and if it less than the overall network density value, the value is 0; thus, the interblock density matrix and image matrix are obtained, as shown in Table 4. The calculation results in Table 4 reflect the relationship between the four blocks. The R-squared is 0.434. The spillover effect of Block 1 is mainly reflected in Block 1, but a spillover effect from Block 1 to Block 3 also exists. The spillover effect of Block 2 is mainly reflected in Block 2, but a spillover effect from Block 1 to Block 4 also exists. The spillover effect of Block 3 is reflected in Block 1 and Block 2. The spillover effect of Block 4 is also reflected in Block 1 and Block 2.

The analysis results of the block model basically verify the current basic situation of regional medical and health development in China. Generally, provinces with high levels of economic development and good resource endowment, such as Beijing and Tianjin, have become favored provinces for patients or those seeking medical care. However, provinces with low levels of economic development (which are mainly concentrated in the central and western regions) have significant spillover effects, and patients or people seeking good medical care have a tendency to flow outward. The corresponding provinces in the Pearl River Delta and Yangtze River Delta, such as Shanghai and Guangdong, not only obtain health economic benefits from other provinces but also spill out, which has a two-way effect.

## 4. Discussion

In the present study, we investigated the possibility of patients moving to resource-dominant regions due to the unequal distribution of medical resources in different provinces. Our results demonstrated that different types of provinces have benefits and spillover effects. Several other studies in China also found that such resources are concentrated mainly in large hospitals located in large or medium-sized cities, which are difficult for patients in rural and remote areas to access [28,29]. Absolute inequalities of health resources increased, while that of health utilization remained constant following China’s health care reform [30]. China is a developing country with a massive population and still faces many challenges in the distribution of regional medical resources, including the uneven distribution of medical resources and an imbalance between supply and demand [31]. As a result, imbalances in regional medical resources may lead to patient mobility, and this mobility has gradually shifted toward central cities in eastern provinces, increasing the pressure on these cities.

In fact, some articles are devoted to the economic, social, and environmental drivers of health inequities, but many of the inequities in health in most regions are determined by factors beyond the health sector [32,33]. Different geographical locations, economic development status, health service talents and service facilities, and even local transportation convenience and national support have all affected the development of regional health services. In China, quality resources are increasingly concentrated in hospitals in developed provinces [34,35]. Excessive resource concentration causes patients with lower medical resource endowments to search for high-level medical services to meet their needs. It should be noted that cross-regional medical treatment is mainly sought for incurable diseases or major diseases. If a local tertiary hospital is unable to treat the disease, the patient migrates as a last resort. Faced with patients with severe or intractable diseases and/or limited financial resources, local hospitals have limited resources and manpower and are unable to effectively treat these diseases or to develop effective solutions. This phenomenon is particularly obvious in areas where medical resources are relatively lacking. Medical equipment, the supply of medicines, the rehabilitation environment, etc., and the lack of important medical personnel involved in treatment cause patients to seek more skilled doctors for treatment. Therefore, cross-regional medical treatment is becoming increasingly frequent. Patients in areas with insufficient medical resources lack trust in local medical standards and are willing to pay more in terms of cost and time to seek medical treatment elsewhere. However, the direct consequence of this phenomenon is the sharp increase in pressure for medical treatment in provinces with medical resource advantages, as shown in Table 5. Although provinces with high endowments of medical resources have relatively smaller populations, they have higher bed utilization rates and higher daily burdens of consultations among physicians. The data also indirectly reflect the unevenness of regional medical resources and high medical pressure in provinces with rich medical resources.

Based on the above discussion, combined with the results of the data analysis, it is not difficult to observe the benefits and spillover effects of the inter-regional health economy spatial network structure. From the perspective of spatial distribution, each block has obvious geographical boundaries that divide the spatial and geographic pattern of the health economy. First, the expansion of the floating population in central cities has increased the work intensity of medical institutions, which is why there is a high rate of bed use and average daily burden of doctors despite a low population in these regions. Second, considering geographical proximity, the provinces with medical resource centers promote medical services in the surrounding provinces and accelerate the concentration of patients from the surrounding provinces to the central provinces. Additional resources are concentrated in these regions, encouraging patients to seek high-quality services [36], as seen from the relationship among Block 1, Block 3, and Block 4. Next, we also find that provinces with spillover effects have to rely on central provinces to improve their health services. However, Block 3 is different from Block 4 in that although Block 3 includes the central provinces, it also features medical institutions with higher medical service levels, so there is a possibility of medical service cooperation. Block 4 not only lacks medical resources but also has insufficient supply in various aspects and cannot provide the same health services as other provinces. Finally, Block 2 includes provinces with relatively abundant medical resources, which not only allows for self-sufficiency but also leads to important spatial relationships with other provinces; this block indirectly supports provinces with underdeveloped medical resources.

It is also found from the analysis of CONCOR that the distribution of medical resources has obvious geographic features. The difference between eastern and western China is stark. Similar results were also found in previous studies [7,37,38]. However, we also find a clear difference between the south and the north. The distribution of medical resources in the north is absolutely concentrated, indicating that there is a large difference in the distribution of resources in northern provinces. This difference has also widened regional health inequality. In contrast, medical resources are more evenly distributed in the south. The gap in medical resources between provinces is relatively small, making health inequality less obvious. Different from the characteristics of concentrated economic development in northern China, the southern region, especially the provinces along the southeast coast, has relatively balanced medical resources because of its relatively balanced economic development.

It is significant that the spatial pattern of China’s health economy is unbalanced. The concentration of medical resources has led to more patients moving towards certain areas. To cope with this situation, local governments must invest more human and financial resources to meet the needs of patients. Ultimately, where resources are more abundant, resources are increasing more quickly. In contrast, where resources are lacking, it is difficult to obtain superior resources. Therefore, the difference in the distribution of medical resources causes an imbalance in the supply and demand of medical services, which indirectly leads to health inequality. However, unlike other industries, health services are related to the lives of individuals, and their fairness is very important in this field. Regional equilibrium reduces the frequency of patient flow and medical costs and ultimately, improves the fairness of resource allocation. Given that reducing health inequities is a key tenet of public health [39], addressing inequality in regional health resources is also a major task. Therefore, we must rely on government policy tools to change the allocation of regional health resources.

This study has several policy implications. First, more health resources, especially medical talent and equipment, are leaning towards resource-poor areas to narrow the capacity gap between different provinces. Second, the existing medical economic space structure should be changed through central financial support. However, relying entirely on financial support will create a huge burden for the government. Other entities, including enterprises and social organizations, should also be added to jointly improve the level of medical treatment in areas with insufficient health resources. Third, cooperation between medical services in different regions should be strengthened and counterpart support to underdeveloped health services should be provided. Specific forms of cooperation include referrals, telemedicine, voluntary services, and medical research, among others. Finally, patients themselves should choose medical institutions rationally based on their health status rather than blindly transferring outward. Governments, hospitals, and other health management agencies should also actively educate the public on how to seek health care.

There are some limitations to this study. This study analyzes the spatial correlation of the health economy in 2017 and shows the geographical characteristics of the distribution of medical resources in China at that time. It would be interesting to perform further analyses on the characteristics of longer term changes and development trends in the distribution of regional medical resources when data are made available. Future research should also focus on analyzing the time course of the development of the health economic spatial network structure. However, the choice of indicators was limited in this study. Although the indicators are consistent with those used in other studies, they may not be comprehensive enough to reflect the entire picture of inequality in health resources and health services. For example, we do not look at the problem of medical treatment in different places from the perspective of individual differences. Relatively high-income families or individuals have the freedom to choose a location, but this does not mean that the province in which they originally lived has poor medical treatment. Unfortunately, the collection of these indicators is difficult, especially because the definition of patient mobility is still insufficient, and individual analyses should be more careful.

## 5. Conclusions

In conclusion, this study demonstrates that significant inequality in the geographic distribution of health resources is evident, despite significant improvements in the level of health services in each region. The benefits and spillover effects of the spatial network structure of the health economy indicate the basic pattern of medical resource allocation. At present, the network structure of China’s health economy is irrational, and resources are concentrated in economically developed provinces. This irrational structure has widened the gap in the level of medical services between provinces. The imbalance in the distribution of medical resources has led to the flow of patients between regions, but this flow has put considerable pressure on resource center cities. Overall, in China, narrowing the gap between regions and achieving resource balance is imperative. Effective health and policy interventions are needed to reduce these inequities and ensure that people in different provinces enjoy the same health service environment. We must find feasible and effective policy solutions, which marks an important step towards achieving coordinated regional development.

Currently, China’s medical system reform is still in progress, and effective solutions are constantly being proposed to solve systemic problems. In the future, the government will work hard to narrow the regional gap to equalize the distribution of regional medical resources. The purpose of this balancing is to serve patients and alleviate the increase in medical costs and health risks caused by large-scale mobility. The government should continue to create a good medical environment for patients. Therefore, the government should make public policies to solve the regional imbalance of medical resources. For example, the government could adopt regional cooperation to help residents in areas with underdeveloped medical resources enjoy the same medical conditions. At the same time, the government should formulate incentive policies to make talents and resources be inclined to the underdeveloped areas of medical resources. These measures can reduce the cross-domain flow of patients and shorten the medical distance. 

## Figures and Tables

**Figure 1 ijerph-18-01096-f001:**
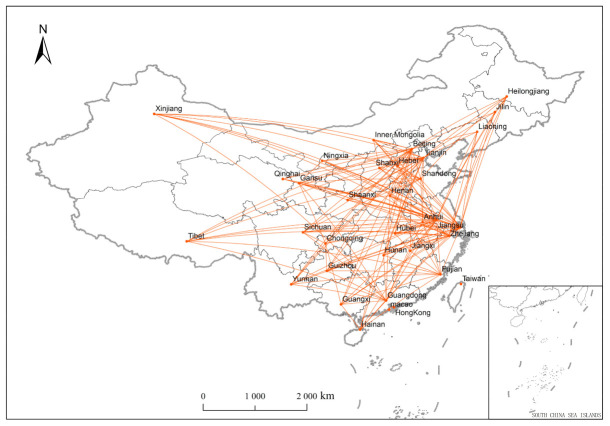
The structure of the network of health economy (2017).

**Figure 2 ijerph-18-01096-f002:**
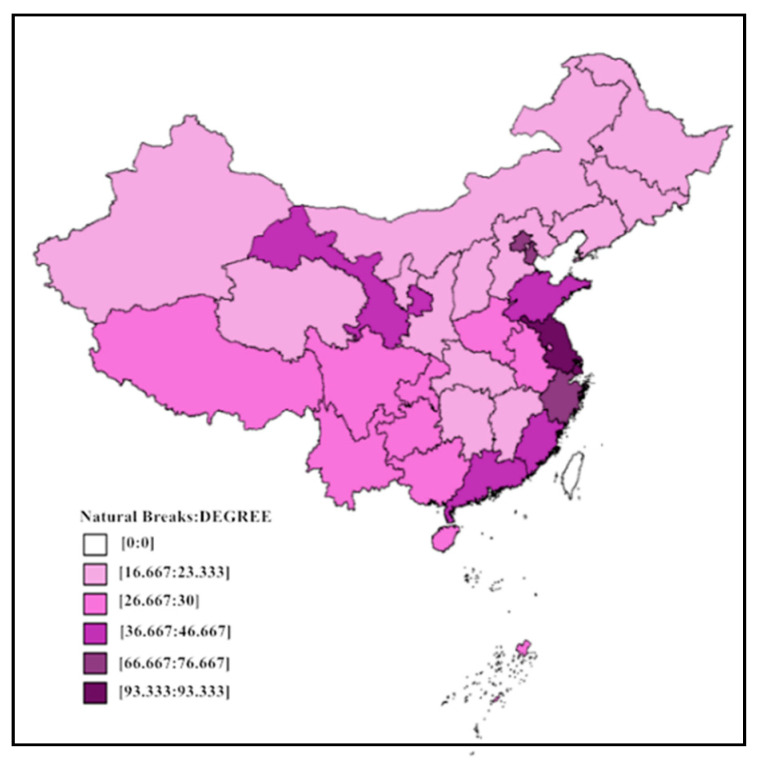
The degree centrality of the health economic spatial network.

**Figure 3 ijerph-18-01096-f003:**
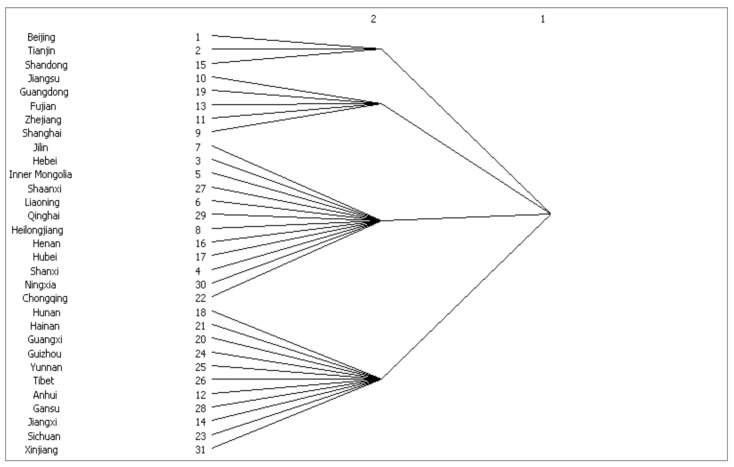
The block health economic spatial network.

**Table 1 ijerph-18-01096-t001:** The network structural characteristics of medical resources.

	Network Size	Number of Relationship	Density	Connectedness	Hierarchy	Efficiency
Structure	31	209	0.2247	1.0000	0.4541	0.6920

**Table 2 ijerph-18-01096-t002:** The network centrality of the health economic spatial network.

Province			Degree Centrality			Closeness Centrality		Betweenness Centrality	
	Out Degree	In Degree	Degree	Benefit or Not	Centrality Rank	Degree	Centrality Rank	Degree	Centrality Rank
Shanghai	9	27	93.333	Yes	1	93.750	1	15.051	1
Jiangsu	4	28	93.333	Yes	1	93.750	1	15.051	1
Beijing	5	23	76.667	Yes	2	81.081	2	10.972	2
Tianjin	5	22	76.667	Yes	2	81.081	2	10.811	3
Zhejiang	5	19	66.667	Yes	3	75.000	3	6.608	4
Guangdong	11	10	46.667	No	4	65.217	4	2.286	5
Shandong	8	11	43.333	Yes	5	63.830	5	1.605	6
Gansu	11	3	36.667	No	6	61.224	6	0.980	8
Fujian	7	7	36.667	Balance	6	61.224	6	1.204	7
Henan	6	9	30.000	Yes	7	58.824	7	0.508	9
Guangxi	7	4	30.000	No	7	58.824	7	0.393	12
Chongqing	8	4	30.000	No	7	58.825	7	0.362	13
Yunnan	8	2	26.667	No	8	57.962	8	0.347	14
Tibet	8	0	26.667	No	8	57.692	9	0.335	15
Sichuan	8	2	26.667	No	8	57.692	9	0.294	16
Hainan	8	1	26.667	No	8	57.692	9	0.235	18
Guizhou	8	2	26.667	No	8	57.692	9	0.347	14
Anhui	3	8	26.667	Yes	8	57.692	9	0.403	11
Xinjiang	7	0	23.333	No	9	56.604	10	0.190	19
Shaanxi	7	1	23.333	No	9	56.604	10	0.106	21
Jiangxi	7	6	23.333	No	9	56.604	10	0.256	17
Hunan	7	3	23.333	No	9	56.604	10	0.235	18
Hubei	7	3	23.333	No	9	56.604	10	0.461	10
Heilongjiang	7	1	23.333	No	9	56.604	10	0.149	20
Qinghai	6	1	20.000	No	10	55.556	11	0.077	23
Jilin	6	1	20.000	No	10	55.556	11	0.077	23
Inner Mongolia	6	1	20.000	No	10	55.556	11	0.089	22
Shanxi	5	4	20.000	No	10	55.556	11	0.089	22
Hebei	5	5	20.000	Balance	10	55.556	11	0.089	22
Liaoning	5	0	16.667	No	11	54.545	12	0.077	23
Ningxia	5	1	16.667	No	11	54.545	12	0.050	24
Average	6.742	6.742	35.269			62.106		2.232	

**Table 3 ijerph-18-01096-t003:** Spillover effects between blocks.

	Receiving Relations		Send Out Relations		Expected Ratio (%)	Actual Ratio (%)	Characteristics of Blocks
	Inside	Outside	Inside	Outside			
Block 1	4	52	4	14	6.667	22.222	Net beneficial block
Block 2	8	83	8	28	13.333	22.222	Bidirectional spillover block
Block 3	5	26	5	68	36.667	6.849	Broker block
Block 4	1	30	1	81	33.333	1.219	Net spillover block

**Table 4 ijerph-18-01096-t004:** Density matrix and image matrix of all blocks.

	Density Matrix				Image Matrix			
	Block 1	Block 2	Block 3	Block 4	Block 1	Block 2	Block 3	Block 4
Block 1	0.667	0.133	0.278	0.061	1	0	1	0
Block 2	0.067	0.400	0.133	0.345	0	1	0	1
Block 3	0.778	0.517	0.038	0.068	1	1	0	0
Block 4	0.697	0.909	0.061	0.009	1	1	0	0

R-squared = 0.434.

**Table 5 ijerph-18-01096-t005:** China’s population and health services by province (2017).

Province	Population (10,000 persons)	Bed Utilization (%)	Burden of Consultations (per day)	Province	Population (10,000 persons)	Bed Utilization (%)	Burden of Consultations (per day)
Guangdong	11169	84.0	10.6	Heilongjiang	3789	78.9	4.7
Shandong	10006	83.4	5.9	Shanxi	3702	77.6	4.2
Henan	9559	88.4	6.1	Guizhou	3580	79.9	5.7
Sichuan	8302	91.3	7.0	Chongqing	3075	84.1	7.2
Jiangsu	8029	87.5	8.7	Jilin	2717	77.6	5.0
Hebei	7520	83.7	5.2	Gansu	2626	81.6	6.2
Hunan	6860	85.2	4.6	Inner Mongolia	2529	74.7	5.1
Anhui	6255	86.2	6.2	Xinjiang	2445	85.0	5.8
Hubei	5902	92.7	6.9	Shanghai	2418	95.4	14.8
Zhejiang	5657	89.4	11.4	Beijing	2171	82.4	9.3
Guangxi	4885	87.7	7.8	Tianjin	1557	78.1	10.4
Yunnan	4801	83.2	7.6	Hainan	926	81.1	6.4
Jiangxi	4622	85.8	5.9	Ningxia	682	80.8	6.8
Liaoning	4369	82.0	5.3	Qinghai	598	70.6	5.3
Fujian	3911	83.1	8.6	Tibet	337	72.1	5.9
Shaanxi	3835	83.7	6.0				

## Data Availability

China Statistical Yearbook: www.stats.gov.cn/tjsj/ndsj. China Health Statistics Yearbook: https://data.cnki.net/area/Yearbook/Single/N2019030282?z=D09.

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
