# Peer review of "Regional Disparity and Patients Mobility: Benefits and Spillover Effects of the Spatial Network Structure of the Health Services in China"

_ijerph, 2021, doi:10.3390/ijerph18031096_

Round 1
Reviewer 1 Report
I found the paper an interesting one, however some improvements might be made to increase the journal's readers experience.
The paper focus on China and once defined the aim and the rationale it straight address the context and the methodology. Providing a brief description of the Chinese health system might be of help to the readers to better understand the discussion and the results' implication.
Some more references might be inserted to support the metrics and the methodology used.
In terms of contributions, the authors might reflect on the replicability of the proposed methodology in other countries context and support the health reforms.
Author Response
Dear reviewer,
Thank you!
It is absolutely our honor to receive your review and valuable comments!
Thank you for your constructive opinions, very helpful for us to improve our research!
All modified text parts are highlighted in the revision paper.
Please see the attachment.
Time is limited and may not reach the acme of perfection.
If there is anything that still I can do, please don’t hesitate to let me know!
Wish you all the health and happiness forever!!
Yours sincerely,
Kaibo Xu
Corresponding author
tdxukaibo@126.com

Reviewer 2 Report
Comments
- The contribution to the (international) literature should be stated in the revised introduction.
- Are there earlier empirical studies using Chinese data on the issue?
- The resolution of figures should be improved.
- The location of the regions should be shown in Appendix.
- The identification assumptions of the model should be stated.
- The concluding section of the paper could provide more practical policy implications that stem from the results that are presented in the paper.
Author Response
Dear reviewer,
Thank you !
It is absolutely our honor to receive your review and valuable comments!
Thank you for your constructive opinions, very helpful for us to improve our research!
All modified text parts are highlighted in the revision paper.
Please see the attachment.
Time is limited and may not reach the acme of perfection.
If there is anything that still I can do, please don’t hesitate to let me know!
Wish you all the health and happiness forever!
Yours sincerely,
Kaibo Xu
Corresponding author
tdxukaibo@126.com

Reviewer 3 Report
Congratulations to the authors of the article, full of enough content to serve the health authorities to rethink the health policy of the country, as should happen in many other places.
It is a novel article because of its methodology and very clear in terms of results and conclusions. Congratulations.
Author Response

(The authors gave the same response as above.)
